# On the Rocky Road to Independence: Big Five Personality Traits and Locus of Control in Polish Primary School Students during Transition into Early Adolescence

**DOI:** 10.3390/ijerph18094564

**Published:** 2021-04-25

**Authors:** Sara Filipiak, Beata Łubianka

**Affiliations:** 1Institute of Psychology, Maria Curie-Skłodowska University in Lublin, Głęboka 45, 20-612 Lublin, Poland; 2Department of Psychology, Jan Kochanowski University in Kielce, Krakowska 11, 25-029 Kielce, Poland; blubianka@ujk.edu.pl

**Keywords:** late childhood, early adolescence, personality traits, locus of control, primary school, changes in the education system

## Abstract

This article reports the results of a survey of 455 Polish primary school sixth-graders experiencing changes in the education system. The goal of the study was to identify the relationships between the Big Five personality traits, measured with the picture-based personality survey for children (PBPS-C) and locus of control, determined using the locus of control questionnaire (LOCQ). The results lead to the conclusion that primary school students do not have an established locus of control of either success or failure. There are also no significant differences between boys and girls in the way they interpret the causes of situations and events that happen to them. Boys, compared to girls, scored significantly higher on traits related to seeking and enjoying the company of others. On the other hand, girls exhibited significantly higher levels of traits responsible for increased anxiety than boys. The personality traits that correlated the strongest with locus of control were Conscientiousness, Openness to Experience, and Agreeableness. A regression model showed that locus of control of success was significantly affected by two traits: Extraversion and Conscientiousness. Locus of control of failure was significantly predicted by Extraversion, Openness to Experience, Agreeableness (positively), and Neuroticism (negatively). Regression model with gender as a moderator of relationships between personality traits and locus of control turned out to be insignificant.

## 1. Introduction

The aim of this study was to analyse personality traits and locus of control of students in the last (sixth) grade of primary school, who were entering adolescence. At the time of the study, the participants were the first generation of students taught in the new education system refashioned by the new education reform in Poland [1]. Before 2017, the Polish system of primary and secondary education had been divided into three levels: six-grade primary schools, three-grade middle schools, and three-grade secondary schools. In 2016, primary school education was extended from six to eight years, and since that time, until 2019, middle schools have been being phased out. As research has repeatedly shown [2], experiences of social change in children’s living environment, including those related to the transition to a next stage of education, are associated with increased stress induced by new requirements and the need to adjust to a new peer group and new teachers [3,4]. Stress resulting from the transition to another stage of education may be additionally aggravated by the hardships of adolescence. The moment of passage from the safe period of childhood to adolescence is connected with numerous challenges related to biological maturation and new social roles and expectations [1,4,5]. Some early adolescents are reluctant to abandon the familiar stability and security of childhood, on the one hand, but on the other, they excitedly enter the more uncertain, more complex, and definitely more dangerous world of adolescence. Emotionally speaking, this is a very demanding process. The curiosity of the new and the growing need for independence motivate young people to take up new challenges on a completely unknown ground. At the same time, however, the uneasiness and sometimes even fear of giving up the comfort of parental care and not having to participate fully in social life makes it difficult for them to part with childhood [6]. Early adolescence is a very important moment in everyone’s life. Students are compelled to cope with the pressures and possibilities associated with becoming an adolescent. The physiological changes occurring in young people’s bodies affect their self-image and may lower their self-esteem and mood [7,8,9]. This is largely due to the neurohormonal changes associated with the pubertal growth spurt. The emotional swing adolescents go through is a consequence of the imbalance between the heightened activity in subcortical emotional processing systems and the immaturity of the prefrontal cortex (PFC) which regulates self-control [10]. For this reason, individuals who enter adolescence have ambivalent emotions, easily go from one extreme to the next, and show exaggerated emotional reactions that are often inappropriate to the context. Puberty opens the world to students in many alluring but also threatening ways [11]. During adolescence prominent developmental transformations are seen in the PFC and limbic brain regions, alterations that include an apparent shift in the balance between mesocortical and mesolimbic dopamine systems. Developmental changes in these stressor-sensitive regions, which are critical for attributing incentive salience to different events and stimuli, likely contribute to the unique characteristics of adolescence [12,13].

Students in early adolescence are at risk of experiencing impairments in personality development [14] and show increased vulnerabilities in behavioural and psychiatric morbidity [5,15]. The period of entering adolescence is also a period of rapid physical growth, the onset of sexual maturation, increase in risk taking and sensation seeking, increased self-consciousness, early experimentation with substance use, and growing rates of accidents, reckless behaviour, depression, anxiety, and eating disorders [12,13,15,16]. It has been claimed that the roots of many adult-onset health problems can be traced back to patterns of behaviour that can be identified in adolescence [5]. Therefore, the period of transition into adolescence reveals complex issues that have enormous implications for public health. Given this background, we decided to survey students in the last grade of primary school who experienced changes in the education system in Poland.

Adolescence is a period of an intensive development of personality, including locus of control, which allows young people to interpret the events in which they participate in their everyday lives and assign meaning to them. Personality traits and locus of control play an important role in predicting successful completion of compulsory education, which, in turn, paves the way for future professional success [17,18,19]. It is important to note that there are also other variables which are crucial for educational achievements and successful adaptation to new demands and roles. Among many, there is evidence that parental involvement, teacher support, and general cognitive abilities [20], as well as goal orientations, popularity, and learning strategies [21], are important factors that shape personal adjustment and educational outcomes of early adolescents. Individual history of so-far education and negative life events also play a crucial role in successful adjustment and the way how adolescents realize their educational duties [22].

By measuring personality traits and locus of control in students, one can partially estimate the level of their adaptation to the new situation and the way they interpret the events they experience [23]. The literature on the subject shows that some personality traits importantly affect the ways early adolescents cope with stress and solve everyday problems. For instance, individuals who are high on Neuroticism have an increased tendency to ruminate about unpleasant events and avoid new and unpredictable situations. By contrast, persons with high levels of Extraversion are more likely to use active problem-solving strategies and interpret new situations as challenges rather than threats [24,25,26]. As for locus of control, Borecka-Biernat [4] has observed that early adolescents who have an internal locus of control are more likely to use active strategies for solving social conflicts.

### 1.1. Personality Traits in Early Adolescence

Currently, one of the most popular personality trait models used in psychological studies of children and adolescents is McCrae’s and Costa [27,28,29] Big Five model. The authors of this model define personality as a dimension of individual differences relating to traits that are responsible for the characteristic ways people feel, think, and behave. According to Costa and McCrae, personality traits are internal dispositions that mould people’s self-image, sense of self-efficacy, and personal goals [30]. They identify five universal dimensions of personality: Extraversion, Neuroticism, Conscientiousness, Agreeableness, and Openness to Experience [27,28]. Persons with high levels of Extraversion are sociable, spontaneously show their emotions, and easily establish new interpersonal relationships. Individuals who are high on Neuroticism have mood swings, feel anxious, often for no reason, and use defensive coping strategies. Conscientious people like order, are well-organized, hard-working, and capable of postponing gratification. Agreeable individuals are compassionate and caring, avoid conflict, and strive to have harmonious relationships with other people. Persons who are open to experience show mental flexibility, cognitive curiosity, versatile interests, and readiness to analyse the complex determinants of their own and other people’s decisions and emotions [28].

Costa and McCrae believe that personality traits are biologically determined and are manifested in the ways even very young children respond to various situations. Personality is strongly influenced by the functions of the PFC (Conscientiousness and Openness to Experience) [31] and by motivation systems that organize responses to rewards and punishments and that drive approach and avoidance behaviour (Extroversion, Neuroticism and Agreeableness) [32]. Bouchard and Loehin [33] also argue that the Big Five traits are biologically based, indicating that the genetic factor explains about 50% of the variance in the Big Five personality traits. Temperamental traits observed in early childhood, such as positive versus negative affect, level of activity, or effortful control, allow one to predict the development of the Big Five traits in later stages of childhood and adolescence [34,35,36]. Costa and McCrae note that personality development is not complete until the early thirties [37], as personality may change due to factors apart from intrinsic maturation [38]. Researchers note that personality traits are also an effect of socialization [39,40,41] and may alter in certain periods of life, especially under the influence of changes in the external environment, assumption of new social roles, teaching, upbringing, and significant life changes. It has been pointed out that the moment of transition from childhood to adolescence abounds in changes in various spheres of young people’s functioning (both biological and socio-emotional), which lead to vital changes in personality [42,43].

Despite the fact that personality traits are most often analysed in older adolescents and adults, existing findings show that studies of younger children are also warranted. For middle childhood, personality has been considered a potent predictor of concurrent and later adaptation [44]. Already Freud noted that personality traits formed in late childhood exerted an influence on personality adjustment in later developmental periods, including adolescence and adulthood [45]. In this light, it is not surprising that personality traits of ever younger individuals are studied more and more often [44]. Nonetheless, the literature on developmental changes in the Big Five personality traits in younger children is scanty [46].

In a study by Suldo, Minch, and Hearon [47], all Big Five personality traits were significant predictors of life satisfaction in adolescents, with the reservation that Agreeableness was associated with life satisfaction only in girls. Very similar findings were obtained by Weber and Huebner [48], who studied 344 American early adolescents. In this group of teenagers, Neuroticism correlated negatively—and Extraversion, Agreeableness and Conscientiousness positively—with general life satisfaction. In addition, Marcionetti and Rossier [49] showed that Conscientiousness, Extraversion, and Neuroticism were associated with life satisfaction in Swiss adolescents. Additionally, they demonstrated that satisfaction with life was connected with high self-efficacy and high self-esteem.

Today, personality psychologists are particularly interested in the developmental changes in the Big Five personality traits occurring during adolescence [50,51,52]. Pullmann, Raudsepp, and Allik [51], who conducted a longitudinal study in a sample of 2650 Estonian adolescents aged 12–18 years, observed the greatest developmental changes in personality traits in the youngest group of respondents (between 12 and 14 years of age). During the 2 year study, the researchers recorded a significant increase in the level of Extraversion in girls and a decrease in the levels of Neuroticism and Agreeableness in both genders.

Research also shows that personality traits are associated with intelligence, school achievement, adjustment, coping, willingness to engage in risk-taking behaviours, and social competence. Graziano and Ward [53], who studied a group of early adolescents, showed that Conscientiousness played an important role in successful school adjustment. Gullone and Moore [54] found that in a sample of Australian teenagers aged from 11 to 18, Extraversion and Agreeableness were negatively associated with the willingness to take risk. Heaven and Ciarrochi [17], who analysed relationships of personality traits and intelligence with school achievement among Australian adolescents, additionally showed that Openness to Experience was a good predictor of school achievement, though only in respondents with higher levels of intelligence. Medvedova [55] observed that, in early adolescents, Neuroticism correlated positively with avoidance coping, and Extraversion was positively associated with support seeking and task-oriented coping. Shiner [44] demonstrated that Extraversion and Agreeableness were predictors of social competence, pro-social behaviour and peer acceptance in 10 year olds.

### 1.2. Locus of Control in Early Adolescence

Late childhood and early adolescence are considered to be crucial for the development of action orientations and for conceptualisations of person–environment transactions [56,57]. For this reason, locus of control develops intensively in these periods of life. Locus of control in Rotter’s theory of social learning [58] is the degree to which people believe they have control over the causes of various events they experience in their lives [59]. Locus of control is a person’s belief about their ability to control the course of events in which they participate [60]. The theoretical construct of locus of control is a continuum that runs from internal to external. People with a strong internal locus of control believe that they have control over both positive and negative events in their lives and that these events are primarily a result of their own actions, features, and competence. They think they have influence on the course of events in their surroundings and feel responsible for their decisions. By contrast, people with a strong external locus of control are more likely to believe that events and reinforcements (rewards and punishment) in their lives are due to factors that are beyond their control, such as bad luck, chance, or unfavourable circumstances [58,61].

Locus of control may be different for different spheres of human life. For instance, a person who believes that their social success is due to their own mental characteristics may at the same time think that their educational or occupational failures are caused by bad luck or other people’s bad intentions. Research shows that students with an internal locus of control are less likely to perceive negative events in their lives as being due to external factors. They are more likely to take action to remedy the difficult situation, for example, by looking to others for support [62], which is especially important in the case of students who transition from one stage of education to the next. Locus of control develops in childhood under the educational influences of the family and school, which stimulate the development of a sense of responsibility for the actions one takes [56]. It is also a result of the child’s increasing independence and their growing ability to actively influence events and derive satisfaction from the effects of their actions. One of the most important factors that may induce developmental changes in locus of control is a change of social environment, including a transition to the next stage of education. Rotter’s is a personality-dominant approach to locus of control. Another approach, introduced in 1977 by Stern and Manifold, treats internal locus of control as a social value. The theory of the norm of internality puts emphasis on social judgements which favour internality as a way of interpreting different events [63]. Social superiority of internal students results from the fact that social appraisers in schools prefer them to externals. The norm of internality, as many researchers suggest, is particularly salient in education as it plays an important role in the assessment of children’s academic potential [64,65,66]. According to this theory, preference for internal causal explanations is learnt at schools where children discover that it is a socially desirable norm [66]. Dubois and Le Poultier [67] conducted studies in which teachers had to predict which fictitious students would pass into the next grade. They also obtained information about the social class of the pupils’ families (lower versus higher) and the children’s level of academic achievements (low versus average). The children filled out an internality questionnaire (with an external versus internal orientation), and the teachers had to make predictions concerning their success in passing to the next grade. Despite important cues concerning family status and academic potential, children who gave more internal explanations received more favourable judgements from the teachers.

Chubb, Fertmani, and Ross [68] note that the development of locus of control in adolescence reflects the challenges and difficulties faced by adolescents. These challenges are related to the need to reconcile learning with free-time activities and adapt to the new peer environment when passing to the next stage of education. In a study of early adolescents, Krampen [56] observed that parental approval and attention to the child’s positive behaviour predicted an internal locus of control. Kaya [69] found that locus of control may also depend on a person’s sociometric status. Students classified as popular and controversial were characterized by a significantly more internal locus of control compared to students with other statuses. It also turns out that participation in sport is conducive to the development of an internal locus of control in early adolescence [57]. Kulas [70] studied changes in students’ locus of control in late childhood and early adolescence over several years of their school life. Locus of internal control decreased as students passed from grade three to grade four of primary school and then increased between grades four and seven. The shift towards a more external locus of control may have been associated with the novelty of the situation related to the transition to grade four, in which there is a notable change in the structure of educational requirements. According to Kulas, adjustment to the new situation and the prospect of it remaining stable for the next several years promoted the development of an internal locus of control in the students he surveyed. Many authors also point to the relationship between locus of control in adolescents and such variables as educational achievement [66,67,71], stress coping [72], personal resilience [73], and social support [74]. Butler-Sweeney [72] demonstrated that internal locus of control was associated with the task-focused coping style and higher self-esteem in girls. Cazan and Dumitrescu [73], on the other hand, found that internal locus of control was related to psychological resilience in adolescents. Mendolia and Walker [18] discovered a positive relationship between Conscientiousness and achievement of long-term life goals and a negative association between low self-esteem and an external locus of control in teenage youth. These researchers point out that the organizational effort young people invest in implementing their plans significantly reduces the possibility of them dropping out from school or work. Young people who have a low self-esteem and an external locus of control are more at risk of leaving school or work.

### 1.3. Personality Traits and Locus of Control in Early Adolescence

Although personality traits and locus of control are separate theoretical constructs, they share some common features. One of them is that they allow to predict how an individual will behave in an unknown situation. This knowledge can help professionals plan effective teaching and educational activities for young people. While personality traits are biologically based and determined, the locus of control, which is a construct derived from the theory of social learning, is shaped by external experiences and social requirements. Personality traits, being internal dispositions and tendencies to engage in or withdraw from particular activities and to interpret various situations in specific ways, may secondarily lead to the development, in children and adolescents, of a sense of responsibility (or a lack thereof) for various situations they experience directly. Firstly, personality traits may predispose young people to seek activities and ways of spending time that suit their individual preferences. Secondly, they can also influence the mode in which teenagers interpret reward and failure situations as being within or beyond their personal control [42,43,75]. What is important here is cognitive assessment of the situation with respect to one’s resources and estimation of the probability of success or failure. A person who feels they have the resources needed to complete a task is likely to undertake the task. By contrast, a person who is not so sure about their resources is likely to avoid new and unpredictable situations associated with a risk of failure [76].

Personality is strongly influenced by motivation systems that organise responses to rewards and punishments and that drive approach and avoidance behaviours [36]. Personality traits influence the processes of assigning motivational meanings to various stimuli and situations [77]. Extraverts tend to seek rewarding situations and see new and ambiguous situations as challenges and opportunities for reward [36,78,79]. In addition to looking for and engaging in potentially rewarding situations, extraverted individuals derive satisfaction from their successes longer than do individuals with elevated Neuroticism levels. Neurotics are characterized by increased sensitivity to punishment [75,78,79], and when they fail, they tend to ruminate and have intrusive thoughts [80,81]. Failures do not demotivate extraverts as strongly as they do to neurotics. For this reason, it has been pointed out that people who are extraverted and open to experience cope better with stress and are less likely than people with high levels of Neuroticism to avoid situations that are associated with taking on challenges and a threat of losing positive self-esteem in the event of failure [76,78].

These findings are important from the point of view of children and adolescents’ developing sense of influence over various life events. Getting involved in situations that may potentially bring various benefits can be conducive to developing an internal locus of control. Conversely, avoiding uncertain situations and withdrawing from the risk associated with making decisions in new and ambiguous circumstances may result in the perpetuation of a sense of lack of influence on what is happening and may incite young people to attribute the causes of events to factors that are beyond their control, such as bad luck or unfavourable conditions.

### 1.4. Gender Differences in Personality Traits and Locus of Control

Limura and Taku [82] found, in a study of Japanese teenagers, that girls scored higher on Conscientiousness than boys. Very similar results were obtained by Klimstra et al. [83] in a sample of early adolescents. The girls who participated in their study had higher levels of Openness to Experience than their male counterparts. In Polish studies of adolescents, girls and boys have been observed to differ in terms of the Big Five personality traits. Lickiewicz [84], in a study involving 236 Polish teenagers, found that girls had higher levels of Extraversion, Openness to Experience, and Agreeableness compared to boys. Czerniawska [85], who studied a group of 200 middle school and secondary school students, observed that boys scored higher on Neuroticism than girls, while girls had higher Openness to Experience scores compared to boys. Branje, van Lieshout, and Gerris [50] demonstrated on the basis of a longitudinal study of 285 Dutch adolescents that Extraversion and Openness to Experience decreased with age in boys, while the opposite was true of Extraversion, Agreeableness, Conscientiousness, and Openness to Experience in girls. An intercultural study carried out by De Bolle et al. [86] in a sample of 4850 adolescents aged from 12 to 17 years showed that girls around the age of 14 exhibited a higher level of Neuroticism than boys in the same age group. This is consistent with the results of research on the levels of Neuroticism in adult women and men, in which female respondents obtained higher scores than their male counterparts. In De Bolle et al.’s study, girls also scored higher on Openness to Experience and Conscientiousness than boys, regardless of age. These differences were independent of culture, which suggests that the differences in personality traits between girls and boys may be universal. In his longitudinal study, Soto [87] obtained similar results in a group of teenage girls. Akhtar and Saxena [88] found that adolescent boys were more internal than girls. Similarly, Kulas [70] observed that adolescent girls were characterized by a significantly more external locus of control compared to boys, which confirms the assumption documented in the literature about higher external locus of control rather in women than in men [89]. The shift towards a more external locus of control in early adolescent girls may be due to the psycho-physiological changes associated with the pubertal growth spurt. Young people may interpret the necessity to accept the changes occurring in their bodies as either opportunities or threats. Girls are more likely than boys to see these changes as giving rise to confusion and diminishing their confidence about the activities they engage in [90,91].

### 1.5. The Present Study

The variables analysed in the present article play an important role in the contemporary psychological discussion on the possibility of predicting young people’s educational achievements [18,92,93] and satisfaction with life [47,94,95], as well as shaping their professional careers [19]. A review of previous research on Costa and McCrae’s [27] Big Five personality traits and Rotter’s [58] concept of locus of control allowed us to formulate the following research questions:1.What are the differences in personality trait configurations between primary school girls and boys?2.What are the differences in locus of control of success and failure between primary school girls and boys?3.What is the relationship between personality traits and locus of control of success and failure in primary school sixth grade students?

In order to answer these research questions, we formulated three hypotheses:

**Hypothesis** **1** **(H1).**
*Girls are more likely than boys to have higher levels of Openness to Experience, Neuroticism, and Conscientiousness.*


**Hypothesis** **2** **(H2)****.**
*Girls are more likely than boys to have an external locus of control of success and failure.*


**Hypothesis** **3** **(H3)****.**
*In the whole group, Extraversion, Openness to Experience, Conscientiousness, and Agreeableness are associated with internal locus of control of successes and failures. Neuroticism is associated with external locus of control of successes and failures. It was presumed that gender is a moderator of relationships between personality traits and locus of control.*


Research on differences in personality traits and locus of control between boys and girls is sometimes inconsistent. Given these previous findings, we assumed that the girls and boys participating in the present study would differ in personality traits and locus of control. In light of the research, we also assumed different relationships between personality traits and locus of control as different personality traits may have different influence on the locus of control. Some traits promote internal, while others promote external locus of control in adolescents [78,79,96,97].

## 2. Materials and Methods

### 2.1. Participants and Procedure

The sample consisted of primary school sixth graders from the city of Lublin, Poland. A total of 455 students took part in the study, including 232 girls and 223 boys, min 12 and max 13 years old. The mean age of the respondents was M = 12.43, and standard deviation was SD = 0.50. Surveys were conducted in four primary schools in Lublin with paper–pencil method by authors of this article. Schools were selected randomly from the list of public schools from the website of the Lublin’s Board of Education. Non-public schools were not taken into consideration. Questionnaires were completed in groups during one meeting with each class. In total, students from 29 classes were examined. The study procedure complied with ethical principles regarding confidentiality, anonymity, and voluntary participation. Consent had been obtained from school principals and parents, and school psychologists/counsellors had been consulted before the survey was conducted. The participants were informed about the scientific purpose of the survey and that they had the right to withdraw from participation at any time. The students filled out a personal data sheet, providing their age and gender details, and two self-report questionnaires: the picture-based personality survey for children (PBPS-C) by Maćkiewicz and Cieciuch [98,99] and the locus of control questionnaire (LOCQ) by Krasowicz and Kurzyp-Wojnarska [100]. The participants took 45 min to complete the survey. As a token of appreciation for their participation in the study, each class received feedback via their homeroom teacher (form tutor) regarding their locus of control scores. All subjects gave their informed consent for inclusion before they participated in the study. The present study was conducted in accordance with the Declaration of Helsinki, and the protocol was approved by the Ethics Committee for Scientific Research of Maria Curie-Skłodowska University, Lublin, Poland (No. 29).

### 2.2. Measures

#### 2.2.1. Picture-Based Personality Survey for Children (PBPS-C)

The Big Five personality traits were measured using the PBPS-C by Maćkiewicz and Cieciuch [98]. It is worth mentioning that currently in Poland there are no instruments comparable to PBPS-C for measuring the Big Five personality traits in early adolescents. Different language versions of this test are successfully used to measure personality traits of children and adolescents in many European countries, among others, Italy [101], Germany [102], France [103], Greece [104], and Spain [105]. It was assumed that the children surveyed, who were transitioning from childhood to early adolescence, could use mental operations that allowed them to accurately assess their various personality characteristics by comparing themselves to the personality characteristics of other people [46]. Following Caspi, Roberts, and Shiner [106], Maćkiewicz and Cieciuch assumed that personality traits and their development in children should be measured with a tool that went beyond the classical questionnaire format based on written descriptions of personality traits. Their picture-based survey takes into account the cognitive abilities of children in the periods of late childhood and early adolescence, who have not yet fully developed the ability to think abstractly. PBPSC is comprised of 15 items, each of which consists of two pictures showing the same character behaving in different ways. Above each pair of pictures, there is a brief caption describing a certain situation, e.g., “When it rains”. The participants are instructed to read the caption and then choose the one picture that best represents the way they would behave in this situation: “Think how you usually behave in a situation like that”. The subjects are told that they should think whether, in the given situation, they are more likely to behave like the character in the picture on the right or on the left. Then, the subjects rate on a five-point scale how similar the character’s behaviour is to their own by ticking an appropriate box (rating from 1—“very similar” to 5—“not similar at all”). The respondents’ choices are compared with the scoring key. The overall score for each scale ranges from 3 to 15 points. Because this instrument has no norms, the scores are interpreted by comparing to the theoretical mean M = 9.0 and the theoretical mean standard deviation SD = 2.0. The validity of PBPS-C was determined using confirmatory factor analysis of Sarisa and Gallhofer’s data quality indicator, defined as the product of validity and reliability. The values of this indicator for the individual PBPS-C scales are as follows: Extraversion 0.86; Agreeableness 0.88; Conscientiousness 0.83; Neuroticism 0.76; and Openness to Experience 0.77. PBPSC has a satisfactory data quality indicator value, which means it can be used in scientific research on the Big Five personality traits of children and adolescents [98] (pp. 77–79). The scale is available in two versions: for younger children in grades 1–3 of primary school and for older children in grades 4–6 of primary school. In the present study, the second version of the questionnaire was used. According to Jean Piaget [107], early adolescence involves a shift from concrete to abstract thinking. This means that thinking becomes hypothetical and allows individuals to solve verbal-conceptual problems. It is worth emphasizing, however, that the cognitive changes occur gradually in adolescents, have different pace, and are characterized by interindividual differences (such as socioeconomical status of the family, history of hospitalization or so far educational achievements etc.).

#### 2.2.2. Locus of Control Questionnaire (LOCQ)

The locus of control variable was measured using the LOCQ by Krasowicz and Kurzyp-Wojnarska [100], a Polish version of Rotter’s I-E instrument based on his theory of Rotter’s social learning and locus of control [54]. The LOCQ and I-E differ in the way the scores are interpreted: in the LOCQ, the higher the score, the more internal the person’s locus of control is (reversely to I-E). At the time of the study, in the school year 2016/2017, the LOCQ had been the up-to-date version of the instrument, before the revised version (LOCQ-R) was issued in 2017 [108]. The test measures locus of control of the consequences of behaviour, a construct described in Rotter’s social learning theory. The LOCQ consists of 46 items, 36 of which are diagnostic for locus of control. The minimum and maximum raw scores are 0 and 18, respectively, for the success and failure scales, and 0 and 36 for the generalized locus of control scale. Half of the diagnostic items regard situations of success and the other half—situations of failure; the sum of the scores on those items is a measure of a generalized locus of control, i.e., a general tendency to attribute more internal or more external causes to various events. The higher score, the higher internal locus of control. The respondent is addressed directly and asked about situations he or she has experienced. External factors responsible for success or failure include a wide range of situations associated with parents, peers, school, etc. Internal factors, on the other hand, include the respondent’s own behaviours and efforts [100].

There are two versions of the LOCQ: one for girls and one for boys. They only differ in the grammatical form of the questions (the Polish language has gendered verb and adjective forms); the content of the questions is the same. After reading an item, the respondent chooses one of two alternatives (a or b), the one that best describes his/her own locus of control in the situation depicted by the item. Cronbach’s alpha reliability coefficients for the individual scales are: α = 0.40 (success), α = 0.54 (failure), and α = 0.62 (global). Despite the low values of the coefficients, we decided to use the LOCQ in the survey because at the time, no other tests for measuring locus of control of success and failure in adolescents aged 12–13 years were available in Polish.

The global score is the sum of the scores obtained on the success and failure subscales and is a measure of a generalized locus of control, not differentiated into locus of control in situations of success and failure. It provides information on a person’s generalized tendency to attribute responsibility for various events to internal or external factors. Subscale scores give a more detailed picture of the typical ways a person interprets the causes of events related to successes and failures.

### 2.3. Data Analysis

The data obtained in the survey were interpreted quantitatively on the basis of raw scores for both the LOCQ and the PBPS-C. Statistical analyses were performed using SPSS v.25. First, normal distribution of the scores was tested using the Kolmogorov–Smirnov test. The distribution of the variables did not deviate from the normal distribution. In further analyses, a parametric Student’s *t*-test of significance of differences was used, and the effect size was measured using Hedge’s *g* coefficient. Correlation analysis was performed using Pearson’s *r*, and a regression analysis was carried out in which personality traits were predictors, and locus of control in situations of successes and failures were dependent variables. Moreover, the analysis of interaction effect of gender on relationships between personality traits and locus of control was carried out. Before Student’s *t*-test was applied, homogeneity of variance of data obtained for the first and second research question had been tested using Levene’s F-test. The F-test was statistically non-significant.

## 3. Results

The analyses of the results are presented in the same order as the research questions, i.e., first, the significances of differences are given separately for girls’ and boys’ scores; then, the results of the analysis of correlations between the investigated variables for the entire sample are presented; and finally, the results of the regression analysis, with personality traits as predictors and locus of control as the dependent variable, are shown.

The results of comparisons of significance of differences between personality trait scores in the samples of girls and boys are shown in Table 1.

The data indicates that girls differ statistically significantly from boys in terms of Extraversion and Neuroticism, but the effect size for both these personality traits is small. Girls are more prone to anxiety than boys, while boys are more open to making new friends and spending time in the company of other people. Table 2 shows the respondents’ mean locus of control scores for the success, failure, and global scales.

There are no statistically significant differences in locus of control of success and failure and the generalized locus of control between the boys and the girls surveyed. Both boys’ and girls’ locus of control scores are in the average range. Table 3 shows the results of the analysis of correlations between the Big Five personality traits and locus of control (generalized and related to success and failure) for the entire sample.

All the correlations between the participants’ PBPS-C and LOCQ scores are statistically significant. Extraversion correlates moderately positively with locus of control of success, locus of control of failure, and generalized locus of control. Negative weak correlations are observed between Neuroticism and the three types of locus of control. Openness to Experience correlates moderately with locus of control of success and generalized locus control, and there is also a weak relationship between Openness to Experience and locus of control of failure. Conscientiousness correlates moderately with locus of control of success and generalized locus of control, and weakly with locus of control of failure. The last personality trait, Agreeableness, correlates weakly with locus of control of success and locus of control of failure, and moderately with generalized locus of control. Table 4 shows the results of the analysis of correlations between the Big Five personality traits and locus of control related to success and failure.

Data presented in Table 4 indicate that there is no significant interaction effect for gender and relationships between personality traits and locus of control. Model with interaction effect of gender fits the data *F* (11, 443) = 8.78; *p* < 0.001; however, introduction of gender to the model did not increase percentage of explained variance of locus of control (1%).

Table 5 presents the results of the multiple regression analysis in which personality traits were predictors and locus of control (of success and failure) were dependent variables. The regression model fits the data as follows: locus of control of success (*F* (5, 449) = 17.86, *p* < 0.001) and locus of control of failure (*F* (5, 449) = 17.87, *p* < 0.001). The regression coefficients show that locus of control of success is associated positively with Extroversion, Openness to Experience, and Conscientiousness. Locus of control of failure, on the other hand, is associated positively with Extroversion, Openness to Experience, and Agreeableness, and negatively with Neuroticism. All the coefficients indicate that there is a significant but weak relationship between personality traits and locus of control of success and failure. The model tested explains a low percentage of variance in each personality trait (R^2^ = 17%). The values of test of variance inflation factor (VIF) coefficient for all personality traits are less than 2, which means that there is low relationships between personality traits.

## 4. Discussion

There are no systematic studies on the associations between personality traits and locus of control in students entering adolescence, especially those who experience changes in educational arrangements, for instance, due to an educational reform [74,109]. Given this gap in research, we decided to carry out the study reported in this article.

In the first hypothesis (H1), we assumed that girls were more likely than boys to have higher levels of Openness to Experience, Neuroticism, and Conscientiousness. The results partially confirm this hypothesis. Costa and McCrae [27,28] have demonstrated that women, on average, have higher Neuroticism levels than men. It has been noted, however, that gender differences in this trait appear already in early adolescence [86,87,110], and our study confirms this finding. This is explained in many ways: among other things, by differences in socialization patterns between girls and boys, as well as the influence of culture and social expectations on the roles they play. Therefore, in Western cultures, the differences in Neuroticism between men and women are more pronounced, possibly due to the egalitarian division of roles between women and men. Women, who take on many different life roles, experience more stress [28,110,111,112,113]. As for the hypothesis that girls should score higher than boys on Openness to Experience and Conscientiousness, our results did not confirm it. Thus, they are in contrast to other studies, including those involving Polish youth [83,84,85]. It is worth noting, however, that the discrepancy between the results may be due to the fact that previous studies had been conducted before 2010, with adolescents who had not had the same educational experiences as the present participants, who had to face the challenges posed by the education reform of 2016. Moreover, personality was measured in those studies using different questionnaires, which makes it impossible to compare the results. In this present study, boys were more extroverted than girls. This finding is similar to that obtained by Branje et al. [114]. It is worth noting that, from a theoretical point of view, Extroversion includes an energy component connected with increased activity, positive emotions, and excitement seeking. This energy component can be seen as conceptually opposite to anhedonia, a central feature of depression [86]. It has been pointed out that in early adolescence, girls, compared to boys, are more at risk of having mood swings and depressive symptoms, which is due to neurohormonal changes. These changes may also explain the higher Extroversion levels in young boys [86,115,116].

Additional analyses conducted in reference to the second hypothesis, in which we assumed that girls are more likely than boys to have an external locus of control of success and failure, demonstrated that there were no differences between girls and boys in this trait. This observation is inconsistent with the results reported by Kulas [70] who showed, in a longitudinal study of younger adolescents, that girls, but not boys, experienced a shift towards a more external locus of control. Wade [117] also observed that girls were more external than boys, although his subjects were older adolescents. It is worth noting, however, that our results confirm those obtained by Manger and Eikeland [118], who found no differences in the locus of control between girls and boys aged 14 and 15.

In the third hypothesis, we assumed for the whole group, Extraversion, Openness to Experience, Conscientiousness, and Agreeableness are associated with internal locus of control of successes and failures. Neuroticism is associated with external locus of control of successes and failures. Additionally, we presumed that gender is a moderator of relationships between personality traits and locus of control. The correlation analysis demonstrated that Extraversion was moderately positively correlated with an internal locus of control of successes and failures and generalized control in all the students surveyed, as was expected. Extraverted behaviours associated with the readiness to form interpersonal relationships and derive satisfaction from them contribute to the sense of agency and internal control over the events in which one participates. It has been pointed out that adolescence is a period of intensive development of social relationships [119], and Extraversion may positively influence the formation of an internal locus of control, because it enables gaining satisfaction from peer relationships. Moreover, extraverts are prone to seek rewarding situations. In addition to looking for and engaging in potentially rewarding situations, extraverted individuals derive satisfaction from their successes longer than do individuals with elevated Neuroticism [36,78,79]. In turn, it may lead to the development of self-efficacy and internal locus of control. Moreover, in this present study, weak negative correlations were observed between Neuroticism and the three types of locus of control. This result is consistent with previous studies, which indicate strong interdependence of neuroticism and external locus of control [120,121,122]. Most likely, heightened anxiety, which is a core of neuroticism, and a tendency to worry coexist with a shift towards a more external locus of control, because, when experiencing difficult emotions, people are more likely to resort to strategies protecting their self-image [86,121,123]. For this reason, they may fail to see their personal influence on the various events they participate in, and they are more likely to look for the causes of failure in external factors.

We found that Openness to Experience correlated moderately with locus of control of success and generalized locus control, and weakly, but significantly, with locus of control of failure. Openness to Experience has been demonstrated to promote the use of diverse problem-solving strategies and in-depth analysis of complex causes of events, human intentions, and feelings. The ability to change a strategy of action when it turns out to be ineffective increases the likelihood of success and may secondarily contribute to the development of an internal locus of control of success [96,124]. The ability to analyse the complex causes of failure may, in turn, foster introspection and allow individuals to contemplate the motives or intentions which have led to the unfavourable outcome.

In our study, Conscientiousness was found to correlate moderately with locus of control of success and generalized locus of control and weakly with locus of control of failure. Conscientiousness, which involves the ability to work hard and continue one’s efforts with patience, has been demonstrated to play an important role in the successful completion of compulsory education [125,126]. School is one of the basic areas of activity of young people, which is why educational successes and failures influence their quality of life [127,128,129]. The more conscientious a person is, the more likely they are to achieve success and then reap the benefits associated with it, such as maintaining positive affect, receiving social gratification (praise) from teachers or parents, etc. In this study, Conscientiousness was found to be conducive to developing an internal locus of control in situations associated with the achievement of success. This hypothesis was also confirmed.

It was expected that Agreeableness would correlate with an internal locus of control. The results indicated that Agreeableness, correlated weakly with locus of control of success and locus of control of failure, and moderately with a generalized locus of control. In light of the research, Agreeableness is broadly considered a key facet of adjustment, motivation to maintain positive interpersonal relations, mental health, and socioemotional competence [97]. In a longitudinal study of a sample of early adolescents, Jensen-Campbell et al. [130] showed that Agreeableness was associated with both peer acceptance and friendship. Moreover, it protected children from victimization by peers. Because the quality of peer relationships is an important component of general life satisfaction in young people, agreeable persons may find it easier to function in a peer group and are more likely to develop an internal locus of control [131].

The regression model with personality traits as predictors and locus of control as the explained variable was well fitted to the data and showed that Extraversion and Conscientiousness were important predictors of locus of control of success. On the other hand, locus of control of failure was significantly predicted by Extraversion, Openness to Experience, Agreeableness (a positive predictor), and Neuroticism (a negative predictor). This result is in line with the studies quoted above which indicate that the individual personality traits play a role in the achievement of important educational and non-educational goals, and thus the development of a locus of control in students. Development of an internal locus of control of success is most strongly affected by Extraversion and Conscientiousness. These traits have the greatest impact on the quality of life of young people in two ways: Extraversion is conducive to deriving satisfaction from peer relationships, and Conscientiousness—from educational success. Both these areas are important aspects of young people’s everyday functioning [128,131]. In the case of failures, all traits, except for Neuroticism, were found to promote the development of an internal locus of control. This is probably due to the fact that increased anxiety, negative affect, and rumination do not foster a sense of responsibility for failure events, which may protect individuals against losing self-esteem [80,81]. Further analysis concerning gender revealed that there is no significant interaction effect for gender and relationships between personality traits and locus of control. Gender differentiates some personality traits but not the relationships between personality traits and locus of control.

## 5. Conclusions

The focus of this present study was on the analysis of personality traits and locus of control in students entering the period of early adolescence at a socially important moment of an introduction of a major educational reform in Poland.

The present results may be influenced by factors related to the introduction of the education reform of 2016 and the specific character of Polish educational policy and its social reception by young people. Nevertheless, we believe that they provide universal scientific insights into the psychosocial functioning of early adolescents experiencing changes in the education system. The present analyses have practical implications, as they can help specialists in education and psychologists understand the possible causes of young people’s adjustment difficulties or difficulties in meeting school requirements. They can also help students discover new ways of achieving important goals while fully using their personality dispositions and provide support in planning educational programmes aimed at teaching students responsibility and encouraging active participation in everyday educational and non-educational activities. Students who experience stress and are lost in a new educational situation may feel they have no control over what they experience, which may lead to the development of an external locus of control [74].

The present results can be used to plan measures and interventions aimed at supporting their adjustment to the new school environment and requirements. Education reforms are put in place all over Europe with varied success, as described in the literature [132,133,134,135,136,137,138,139,140,141]. Supposedly, the profile of scores obtained in young people in other countries could be different from the one obtained in this study. However, this only testifies to the potential of the present study and the possibility it provides of comparing the results in a wider context.

## 6. Limitations

An important limitation of our study is the fact that the participants were surveyed in only one city in eastern Poland. Therefore, the findings cannot be generalized to the whole population of students experiencing the changes imposed by the education reform in Poland. No detailed information was collected on the socioeconomic status of the students’ families and variables such as parental education or monthly income. These variables should be controlled for in a future study. Likewise, we did not control for parent–children relations, the current students’ achievement level, their sociometric status in the classroom, or their need for psychological support. These factors are important in successful adaptation to new environment and further educational outcomes. Results presented in this study pertain to two variables which are important in predicting academic success but are not the only ones underlying the quality of this process. In addition to taking into account the abovementioned variables in future research, it would also be interesting to examine students’ self-descriptions of their personality traits and compare them with assessments made by their parents and teachers. Studies of this type are very popular because they give a fuller picture of the personality functioning of young people seen from various perspectives [50,107]. With regard to locus of control, some studies show that it should be analysed in specific domains of young people’s life, such as the school environment or peer relationships [118]. In this present study, both successes and failures included a variety of events associated with the respondents’ school life and out-of-school activity. Future investigations should focus on particular spheres of functioning. Due to the interindividual differences in the changes that take place during early adolescence and the complex conditions of the development of personality traits, there is no universal signpost telling adolescents and their parents what life path to follow. Rather, it is young people themselves who begin to seek their own development path with increasing awareness and in keeping with their preferences and dispositions. Due to the fact that personality’s development is a long-term process, longitudinal research with at least a few waves during early, middle, and late adolescence period are necessary. To obtain a bigger and more accurate picture of these changes, future research should take into account other variables and analyse the relationships between them and the development of personality traits and locus of control in adolescents.

## Figures and Tables

**Table 1 ijerph-18-04564-t001:** Descriptive statistics, Student’s *t*-test, and effect size for boys’ (*n* = 223) and girls’ (*n* = 232) PBPS-C scores.

Personality Trait	Boys		Girls		*t*	*p*	*Hedge’s g*
	*M*	*SD*	*M*	*SD*			
Extraversion	12.02	2.42	11.48	2.96	2.106	0.036 **	0.20
Neuroticism	7.95	2.90	8.96	2.48	−3.999	0.001 ***	0.37
Openness to Experience	9.28	2.67	9.54	2.54	−1.085	0.278	0.10
Conscientiousness	10.14	2.75	9.92	2.67	0.870	0.385	0.08
Agreeableness	11.10	2.45	11.50	2.45	0.869	0.081	0.16

Note: ** *p* < 0.05; *** *p* < 0.001.

**Table 2 ijerph-18-04564-t002:** Descriptive statistics, Student’s *t*-test and effect size for boys’ (*n* = 223) and girls’ (*n* = 232) LOCQ scores.

Locus of Control	Boys		Girls		*t*	*p*
	*M*	*SD*	*M*	*SD*		
Success	12.02	2.84	11.83	2.85	0.696	0.487
Failure	11.80	3.61	11.47	3.43	0.995	0.320
Generalized	23.81	5.61	23.28	5.49	1.013	0.312

**Table 3 ijerph-18-04564-t003:** Results of analysis of correlations between PBPS-C and LOCQ scores for the entire sample (*n* = 455).

Personality Trait		Locus of Control		
		Success	Failure	Generalized
Extraversion	*Pearson’s r*	0.32 **	0.34 **	0.33 **
	*p*	0.001	0.001	0.001
Neuroticism	*Pearson’s r*	−0.23 **	−0.24 **	−0.27 **
	*p*	0.001	0.001	0.001
Openness to Experience	*Pearson’s r*	0.31 **	0.25 **	0.30 **
	*p*	0.001	0.001	0.001
Conscientiousness	*Pearson’s r*	0.32 **	0.17 **	0.27 **
	*p*	0.001	0.001	0.001
Agreeableness	*Pearson’s r*	0.21 **	0.28 **	0.30 **
	*p*	0.001	0.001	0.001

Note: ** *p* < 0.001.

**Table 4 ijerph-18-04564-t004:** Results of regression analysis with gender as a moderator of relationships between personality traits and locus of control in the situations of successes and failures.

Predictors		Locus of Control							
		Success				Failure			
		β	SE	*t*	*p*	β	SE	*t*	*p*
Model 1	Constance		1.12	5.523	0.001 ***		1.38	3.490	0.001 ***
	Gender	0.01	0.12	0.216	0.829	0.03	0.16	0.762	0.446
	Extraversion	0.13	0.05	2.730	0.007 **	0.15	0.06	3.269	0.001 ***
	Neuroticism	−0.08	0.05	−1.676	0.094	−0.11	0.06	−2.293	0.022 **
	Openness to Experience	0.14	0.05	3.015	0.003 **	0.16	0.06	3.396	0.001 ***
	Conscientiousness	0.23	0.05	5.009	0.001 ***	0.04	0.06	0.947	0.344
	Agreeableness	0.08	0.05	1.687	0.092	0.20	0.07	4.183	0.001 ***
Model 2	Constance		1.13	5.457	0.001 ***		1.34	3.634	0.001 ***
	Extroversion	0.11	0.05	2.323	0.021 **	0.13	0.06	2.800	0.005 **
	Neuroticism	−0.06	0.05	−1.311	0.191	−0.11	0.06	−2.274	0.023 **
	Openness to Experience	0.14	0.05	3.045	0.002 **	0.17	0.06	3.577	0.001 ***
	Conscientiousness	0.24	0.05	5.121	0.001 ***	0.05	0.06	0.996	0.320
	Agreeableness	0.08	0.05	1.708	0.088	0.19	0.07	4.114	0.001 ***
	Gender	0.01	0.13	0.318	0.751	0.04	0.16	0.803	0.422
	Gender × Extraversion	−0.08	0.13	−1.792	0.074	−0.03	0.17	−0.559	0.577
	Gender × Neuroticism	−0.01	0.14	−0.173	0.863	0.08	0.17	1.684	0.093
	Gender × Openness to Experience	0.03	0.13	0.613	0.540	0.05	0.16	1.123	0.262
	Gender × Conscientiousness	−0.03	0.13	−0.607	0.544	0.001	0.17	0.002	0.998
	Gender × Agreeableness	−0.06	0.13	−1.239	0.216	−0.003	0.17	−0.065	0.948
		R^2^ = 0.17; ΔR^2^ = 0.01				R^2^ = 0.18; ΔR^2^ = 0.01			

Note: ** *p* < 0.01; *** *p* < 0.001.

**Table 5 ijerph-18-04564-t005:** Results of regression analysis with personality traits as predictors and locus of control as dependent variable.

	Locus of Control									
Personality Traits	Success					Failure				
	β	SE	*t*	*p*	VIF	β	SE	*t*	*p*	VIF
Extraversion	0.12	0.05	2.761	0.006 **	1.10	0.15	0.06	3.346	0.001 ***	1.10
Neuroticism	−0.08	0.05	−1.747	0.081	1.13	−0.12	0.06	−2.475	0.014 *	1.13
Openness to Experience	0.14	0.05	3.010	0.003 **	1.21	0.15	0.06	3.350	0.001 ***	1.21
Conscientiousness	0.23	0.05	5.021	0.001 ***	1.24	0.05	0.06	0.967	0.334	1.24
Agreeableness	0.08	0.05	1.675	0.095	1.14	0.19	0.07	4.123	0.001 ***	1.14
	R^2^ = 0.17		Adj R^2^ = 0.17			R^2^ = 0.17		Adj R^2^ = 0.17		

Note: * *p* < 0.05; ** *p* < 0.01; *** *p* < 0.001.

## Data Availability

The data presented in this study are available on request from the corresponding author. The data are not publicly available due to privacy or ethical restrictions.

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
