# Peer review of "On the Rocky Road to Independence: Big Five Personality Traits and Locus of Control in Polish Primary School Students during Transition into Early Adolescence"

_ijerph, 2021, doi:10.3390/ijerph18094564_

Round 1

Reviewer 1 Report

This manuscript presented an interesting study. Overall, this manuscript is well written. I would like to see how the authors will address a few issues described below.

First, how can the authors justify to use linear regression?  The data were collected from 29 classes and four schools, which have nested data structure and may have high ICC or design effect. Typically, researchers will run multilevel models instead of linear regression and calculate design effect to see the necessity for conducting multilevel models. Since you only have four schools here, you may either ignore the school level.

Second, in the linear regression analyses, have the authors calculated VIF to check if there exists multicollinearity issue? If the five personality traits are highly correlated, then multicollinearity issue will be a concern.

Third, “Extroversion correlates weakly positively …” (Line 499) Since the authors used .30 as the criterion for moderate correlations, so they may want to change “weakly” to “moderately” here.

Fourth, “Additional analyses demonstrated that girls were more likely than boys to have an external locus of control of success and failure.” (Lines 553-554) Based on the analysis table (Table 2), I can only see that there is no gender difference. Please explain what additional analyses led authors to make this conclusion.

One small comment on line 425. Did the authors really mean “The higher score, the higher internal locus of control”? My understanding is that the authors referred to LOCQ in this paragraph, so it should be the opposite.

Author Response

Response to Review 1:

We would like to thank Reviewer for taking the time to read our manuscript and for a detailed revision and helpful advice. We are especially grateful for all the comments, particularly those pointing to the weaknesses of the article. Below we would like to present how we corrected the article in line with these comments:

  1. We used linear regression to assess if personality traits are predictors of locus of control of successes and failures, as some research suggest such relation. Predictors were introduced simultaneously to the model. We calculated VIF (and put additional column to table 5) to check if there exist multicollinearity issue. Table 5 is inserted on page 13.
  2. It was our mistake to report „weak” correlation between Extraversion and corrected the statement. Now the sentence is: „Extroversion correlates moderately positively with locus of control of success, locus of control of failure, and generalized locus of control” (lines 576-577).
  3. Thank you for pointing to the wrong statement that girls were more external in locus of control than boys. The description of table 2 was changed.
  4. Concerning the part Measures, in which we explain scores in LOCQ, there is no mistake. In Polish version of the scale measuring locus of control (which is based on I-E questionnaire of J. Rotter theory), the results are interpreted in a different manner: while in I-E questionnaire, the higher score, the more external locus of control is, in LOCQ reversly: the higher score, the higher internal locus of control is.

Sincerely,

Authors

Reviewer 2 Report

The article presents a good, well-organized introduction, which clearly describes the importance of personality traits and locus of control in adapting to new situations in adolescents. However, it focuses a lot on personality traits as possible predictors of success or failure in adapting to new situations, leaving aside other variables that can be very important, such as previous experiences. I think that for such a long and elaborate introduction it leaves many important questions by the wayside and offers too poor work for its results to be relevant.

As for the rest of the work, it is not clear to me how the data was collected, online or with paper and pencil. This point should be clarified (line 355). It is also interesting to know the minimum and maximum values in the age of the participants, although seeing the standard deviation it does not have to be a very wide range, it is convenient to include these values. Were the primary schools selected randomly or by some sampling method?

Regarding the measures, the response scale of the PBPS-C needs to be clarified. The authors indicate that it is a five-point response scale, but do not indicate the verbal anchors. At least the first and last should be indicated. From very similar to nothing similar? In line 390 a dot is needed before the sentence.

In the analysis section it should be indicated which are the predictors of the regression (or regressions) and which are the dependent variables. We can see it only in the Results section. It is also not indicated whether the regressions were performed separately for boys and girls, and the method of introduction of the predictor variables is not indicated either. Looking at the results, I see that the complete sample has been used. However, in Table 1 with the t tests it is observed that there are differences between both boys and girls in two personality traits. For this reason, the regressions should have been carried out considering gender as a moderating variable, since it clearly interacts with personality traits. By not doing so, the regression with the whole sample can offer biased results. It would be better, and would introduce less error, to carry out a path analysis (using the total scores in the factors) considering gender as a moderating variable. In this model, the five personality traits could be considered as predictor variables of the two types of locus of control and adding the variable gender as moderator of the relationship between locus of control and personality traits. These analyzes can be performed with Amos in SPSS.

Regarding the discussion, since the analyses carried out do not consider the moderating effect of gender, it should be rewritten considering the effect of this variable on the regressions. The correlations only indicate whether it is appropriate to introduce the personality factors in the regression as predictors, they are not so important as to dwell so much talking about them in the discussion. On the other hand, and although it is mentioned in the limitations of the study, using only personality traits as predictors of the locus of control is too superficial to be able to explain the adaptive capacity of adolescents. Despite the analyses are regressions, more emphasis should be placed on the fact that they are more about relationships between variables, as there are quite a few important variables missing that could better explain success or failure in adapting to new situations. It seems to me to be a very limited study considering too few predictive variables of adaptive success or failure in adolescents, and I do not think it provides relevant information.

Author Response

Response to Review 2:

We would like to thank Reviewer for taking the time to read our manuscript and for a detailed revision and helpful advice. We are especially grateful for all the comments, particularly those pointing to the weaknesses of the article. Below we would like to present how we corrected the article in line with these comments:

  1. We admit that Introduction section pertains mainly to personality traits and locus of control as the main variables which were measured in this study but lacks in other psychological characteristics important for a positive adaptation to new educational stage in early adolescence. We fully agree with Reviewer that there are many other factors which are crucial in this process. Such factors may be connected with the so-far experiences (which Reviewer mentions), which may be connected both with the individual history of educational path (e.g. repeating classes due to unfavorable level of educational achievements or familiar circumstances such as change of the place of residence that eventually led to the change of the school). There are also important other psychological variables connected with the way how an adolescent copes with stress (and how he or she uses personal resources in face of difficulties). Others like self-concept, popularity, learning strategies are also important and were not controlled in this study and, as Reviewer noted, this fact has important implications for the results. Therefore, we put additional information about these shortcomings in Introduction section (lines: 84-92) and also in Limitation section (lines: 811-814).
  2. Conerning Revierew’s comments on Participants and procedure section:

- we put additional information that questionnaires were collected with a paper-pencil method in a traditional way, during one meeting with each class. Surveys were conducted by authors of this article (lines 420-421)

- we also put additional information about age range of students (they were 12 and 13 years old) and information about a random selection of public schools which were enrolled in this study (lines 418-423)

  1. Thank you for important issue concerning the response scale of PBPS-C as it was not clarified. We put additional information in line 465 indicating that the scale has 5 points in which 1= very similar to me and 5=not similar at all.
  2. In Data analysis section we highlighted which variables are preditors of the regression (personality traits) and which are dependent variables (locus of control in successes and failures) (lines 531-533) as well as information about the method of entry of predictors to the regression model (it was a simultaneous enter method where all variables are entered at the same time). We calculated regression analysis with gender as a moderator of dependencies between personality traits and locus of control. However gender did not interact with the variables (table 4, page 12).
  3. Discussion has been corrected according to Reviewer’s suggestions. We put more information about gender and information about limitations (the fact that we analysed only personality traits and locus of control) whereas there are also other variables crucial for positive adaptation to new environment and academic achievements during adolescence.

Sincerely,

Authors

Reviewer 3 Report

This paper reports the results of a survey on a sample of twelve year olds. This paper addresses an interesting topic and it is a solid paper. I just have a few minor points.

  1. The authors could more emphasize gender differences in personality traits and in the locus of control, highlighting any contradictory results in the literature and dedicating a separate paragraph to this topic in the introduction section.
  2. The authors reported that “research on differences in personality traits and locus of control between boys and girls is sometime inconsistent” (page 7 line 326), thus it is unclear why the first two research questions of the paper are about gender differences. I think that the third research question is more important than the first two and it should be put at the first place.
  3. The authors could add in the discussion section specific cross references to the three hypotheses.
  4. Please explain why extroversion may positively influence the formation of an internal locus of control (page 12 lines 572-574).
  5. Please cite more studies (see for example Judge et al., 2002) in order to explain why anxiety (thus low extroversion?) was positively correlated to low self-esteem and to external locus of control (page 12 lines 576-578). In any case, it could be useful to emphatize that anxiety is related to neuroticism rather than extroversion in previous research (see for example Vittengl, 2017).
  6. Please take into account that personality development is not complete until the end of the decade of the 20s (see Costa and McCrae, 1994), thus it would be interesting to evaluate in future studies how the five personality factors may vary according to the locus of control.

References

Costa Jr, P. T., & McCrae, R. R. (1994). Stability and change in personality from adolescence through adulthood. In C. F. Halverson, Jr., G. A. Kohnstamm, & R. P. Martin (Eds.), The developing structure of temperament and personality from infancy to adulthood (p. 139–150). Lawrence Erlbaum Associates, Inc.

Judge, T. A., Erez, A., Bono, J. E., & Thoresen, C. J. (2002). Are measures of self-esteem, neuroticism, locus of control, and generalized self-efficacy indicators of a common core construct?. Journal of Personality and Social Psychology, 83(3), 693.

Vittengl, J. R. (2017). Who pays the price for high neuroticism? Moderators of longitudinal risks for depression and anxiety. Psychological Medicine, 47(10), 1794.

Author Response

Response to Review 3:

We would like to thank Reviewer for taking the time to read our manuscript and for a detailed revision and helpful advice. We are especially grateful for all the comments, particularly those pointing to the weaknesses of the article. Below we would like to present how we corrected the article in line with these comments:

  1. We made a separate paragraph (1.4. Gender differences in personality traits and locus of control) in Introduction section to emphasize gender differences in personality traits and in the locus of control (line 338)
  2. We agree that there are many inconsistencies regarding differences between boys and girls in personality and locus of control, but due to the fact that there still are some constistencies as depitcted in the literature, for example, higher neuroticism in adolescent girls than in boys (De Bolle et al., 2015; Soto, 2016) or higher conscientiousness in girls (Klimstra et al., 2009; Limura, Taku, 2018) or higher external locus of control also in girls than in boys (Kulas, 1998; Sherman et al., 1997; Akhatar, Saxena, 2014), we decided to leave the two questions about gender differences. As we also regard the third question (concerning gender) as an important one, we made some additional analysis with gender as a moderator of relationships between personality traits and locus of control (new table 4 on page 12). However, regression analysis with gender turned out to be insignificant.
  3. We corrected the Discussion section according to Reviever’s comment. We discuss the results with reference to the three hypotheses.
  4. We explained why extraversion may influence formation of internal locus of control (lines 695-706). We think that other previous information (in lines 320-338) also justify why Extraversion may faciliate development of internal locus of control.
  5. Thank you very much for your suggestion concerning referenceses (Costa, McCrae, 1994; Judge et al., 2002; Vittengl, 2017) and information which spring from these studies, which are important for the assumptions of our study. Anxiety indeed is a factor influencing low self-esteem and to external self-esteem and should be linked mostly to neuroticism, rather than extraversion. It was not our intention to present anxiety as a trait connected with Extraversion, but it was not stated clearly in the text. We put additional information in lines 702-704.
  6. The Reviewer’s remark concerning the fact that personality development is not complete until the end of the decade of 20s. is a very important one. We did not write about this issue directly in our manuscript. We fully accord with Reviewer’s suggestion as many studies state that personality formation is a long-term process which, during adolescence, is „in progress” and is influences by many factors. Moreover, it lasts for many years after (with new roles, social demands and developmental tasks or non-normative events), which are the sources of personality change after adolescence and even during the whole adulthood. We put additional information in Introduction section (lines 132-134) and in Limitations section (about the necessity to conduct longitudinal analysis due to the long-term proces in personality development) (lines 828-830).

We added new references, including these which were suggested by Reviewer:

Costa Jr, P. T., & McCrae, R. R. (1994). Stability and change in personality from adolescence through adulthood. In C. F. Halverson, Jr., G. A. Kohnstamm, & R. P. Martin (Eds.), The developing structure of temperament and personality from infancy to adulthood (p. 139–150). Lawrence Erlbaum Associates, Inc.

Judge, T. A., Erez, A., Bono, J. E., & Thoresen, C. J. (2002). Are measures of self-esteem, neuroticism, locus of control, and generalized self-efficacy indicators of a common core construct?. Journal of Personality and Social Psychology, 83(3), 693.

Vittengl, J. R. (2017). Who pays the price for high neuroticism? Moderators of longitudinal risks for depression and anxiety. Psychological Medicine, 47(10), 1794.

Sincerely, Authors

Round 2

Reviewer 2 Report

I think I did not explain myself well in the review I did. Personality is not clearly formed at such a young age. If the authors previously found differences by sex in the personality data, the model should show it. In any case, a large group of regressions does not provide adequate information, but rather a model of structural equations using, among others, the sex variable as a moderator of relationships between different variables.
Furthermore, the article does not provide relevant and impactful information to be published in a high impact journal. On the other hand, the regressions with moderating variables are a simple and minimal resource, some of my students are doing more extensive work than this as the final degree project.